

# Potential Impact of Climate Change and Extreme Events on Slope Land Hazard - A Case Study of Xindian Watershed in Taiwan

Shih-Chao WEI[1], Hsin-Chi LI[2], Hung-Ju SHIH[2], Ko-Fei LIU[1]

[1] Department of Civil Engineering, National Taiwan University, Taipei 10617, Taiwan
[2] National Science and Technology Center for Disaster Reduction, New Taipei City 23143, Taiwan

*Correspondence to*: Hsin-Chi LI (hsinchi@ncdr.nat.gov.tw)

**Abstract.** The production and transportation of sediment in mountainous areas caused by extreme rainfall events triggered by climate change is a challenging problem, especially in watersheds. To investigate this issue, the present study adopted the scenario approach coupled with simulations using various models. Upon careful model selection, the simulation of projected
rainfall, landslide, debris flow, and loss assessment were integrated by connecting the models' input and output. The Xindian watershed upstream from Taipei, Taiwan, was identified and two extreme rainfall scenarios from the late 20th and 21st centuries were selected to compare the effects of climate change. Using sequence simulations, the chain reaction and compounded disaster were analysed. Moreover, the potential effects of slope land hazards were compared between the present and future, and the likely impacts in the selected watershed areas were discussed with respect to extreme climate. The results
established that the unstable sediment volume would increase by 28.81% in terms of the projected extreme event. The total economic losses caused by the chain impacts of slope land disasters under climate change would be increased to US$ 358.25 million. Owing to the geographical environment of the Taipei metropolitan area, the indirect losses of water supply shortage caused by slope land disasters would be more serious than direct losses. In particular, avenues to ensure the availability of water supply will be the most critical disaster prevention topic in the event of a future slope land disaster. The results obtained
from this study are expected to be beneficial, because they provide critical information for devising long-term strategies to combat the impacts of slope land disasters.

## 1 Introduction

In recent years, the frequency and magnitude of disasters associated with extreme climate have increased. Climate change and the impacts of disasters are major concerns for scientists, policymakers, and the public (IPCC, 2012). Hence, many researchers
have endeavoured to predict future climate events and explore the potential hazards of extreme weather.

As per statistics provided by the World Bank, Taiwan is situated in a high risk area that is especially prone to slope land disasters (Dilley et al., 2005). Therefore, we focused our attention on slope land disasters associated with climate change. In addition, a watershed was selected to explore the potential impacts of slope land problems caused by extreme climate. Studies have established that under extreme climatic conditions, changes in temperature and precipitation can lead to slope land hazards



such as landslides and debris flow in the local area (Hsu et al., 2011). Although temperature can play a critical role in slope land problems, such as snowmelt and glacier wasting (Chiarle et al., 2007;Rebetez et al., 1997;Stoffel et al., 2014), these problems were not included in this study because Taiwan is located in a low altitude region that does not have any glaciers or receive snowfall. Therefore, precipitation is the most important factor associated with climate change in Taiwan, and it is the

only triggering factor of slope land disasters. Hence, this study exclusively addressed this specific problem faced by the country. In Taiwan, common slope land hazards can be classified into two parts, namely landslides and debris flow. To link these hazards with meteorological properties, the scenario approach has been relied upon in recent years. Some potential effects of landslides have been investigated by studying the differences between current and future scenarios (Buma and Dehn, 1998;Collison et al., 2000;Crozier, 2010). Similarly, the frequency of debris flow has been estimated based on the projected

future rainfall (Rebetez et al., 1997;Stoffel et al., 2014). However, the abovementioned studies have concentrated on the relationship between the meteorological properties of climate change and one of the consequent hazards, such as rainfall-triggered landslides or rainfall-triggered debris flow. However, in slope land problems, the various types of hazard are strongly linked or occur as a chain reaction. For example, landslide mass is one of the triggering factors of debris flow.

Furthermore, slope land disasters result in serious casualties and economic loss. For example, Taipei experienced heavy rainfall

during Typhoon Soudelor in August 2015. The accumulated rainfall over a period of 3, 6, and 12 hours exceeded the 200 year record (Wei et al., 2015). Large-scale landslides and debris flows were triggered within this short period because of the intense rainfall. The regional landslide disasters necessitated the closure of roads and the debris flow from the tributaries increased the concentration of suspended solids in the Nanshi River. According to the Taipei Water Department, the peak turbidity reached 39,300 NTU (nephelometric turbidity unit) at 08:00 hours on August 8, 2015, and it took 42 hours for the value to fall to the

permissible limit of 3,000 NTU. Because of the disaster, water supply was disrupted and the water quality was compromised in the subsequent days in the Taipei area. Therefore, a comprehensive assessment of slope land disasters is a critical issue in Taiwan, especially with the increasing threat of typhoon events.

Based on the given rainfall scenario, the consequent landslide, soil erosion, debris flow, sediment transport, and economic loss can be predicted using relevant theories and simulations based on numerical programs. However, the theories of physical

phenomena such as sediment production and transport process have some discrepancies and combining them is difficult. Therefore, some researchers have started to apply suitable numerical techniques to combine different physical phenomena by linking the output from one model to the input of another model. For instance, Liu et al. (2013) combined landslides, debris flow, and sediment transport to simulate turbidity in a reservoir, based on an assumed scenario. Similarly, Wu et al. (2016b) comprehensively assessed the impacts of landslides, debris flow, flooding, and coastline disasters. In the present study, the

scenario approach was used in a selected watershed. Appropriate models were selected after a survey and comparison. An integrated model was constructed for the study area, and the potential impacts of climate change on slope land disasters were examined.



## 2 Methodology

In slope land hazards, sediment production and transport warrant a great deal of attention. The sequence of events from sediment production to its transport can be regarded as a chain reaction in the watershed. For example, in Taiwan, heavy rainfall triggered by typhoons usually leads to large-scale landslides on slope land. When loose landslide deposits mix with the runoff or dammed lake water, the debris flows downstream along the gully. Hence, the chain reaction of sediment flow cannot be ignored and should be considered in the sediment related discussions of mountainous terrains, especially in an extreme rainfall event. In this process, the various physical mechanisms could be simulated by an effective combination of models, such as rainfall, landslide, and debris flow. Loss assessment must be applied for evaluating the economic impacts of slope land disasters (Liu et al., 2013;Wu et al., 2016b). Detailed discussions on each part are introduced in the following sections.

### 2.1 Rainfall Scenarios

In recent decades, the climate projections for various periods are widely studied using general circulation models (GCM). These models are useful for studying the sequence of events that stem from climate change occurring on a global scale. The frequency of natural hazards, risk assessment, and adaptation strategies can be explored with the help of these models. However, the resolution of GCMs is insufficient to simulate data for further application in hydrology or agriculture at the local level. For example, in typhoon-related rainfall, GCM results cannot provide details on weather patterns in the local area. Furthermore, it is not possible to obtain assessments on a daily or hourly basis. To link the simulations of atmosphere and hydrology, downscaling techniques (statistical or dynamic downscaling) are useful. In Taiwan, these techniques have been applied for climate projection simulation by the Taiwan Climate Change Projection and Information Platform (TCCIP) funded by the Ministry of Science and Technology, and the downscaling dataset is freely available on their official website (TCCIP, 2017). For the data provided by TCCIP, the atmospheric general circulation model 3.2 (AGCM 3.2) developed by the Meteorological Research Institute (MRI), Japan Meteorological Agency (JMA), is used for global climate simulation at a 20 km horizontal resolution (Mizuta et al., 2012). The observed sea surface temperatures are considered as lower boundary conditions by the coupled model's intercomparison project phase 5 (CMIP5). Using the initial boundary conditions specified by the results from MRI-AGCM 3.2, the dynamic downscaling dataset at a 5 km horizontal resolution are simulated by the weather research and forecasting modeling system (WRF) (Skamarock et al., 2008) developed by the U.S. National Center for Atmospheric Research (NCAR). Because errors in the estimation of total rainfall are still found in the WRF results, the quantile mapping method is adopted for bias correction in these datasets (Su et al., 2016).

According to the representative concentration pathway 8.5 (RCP 8.5) scenario issued by the IPCC fifth assessment report (AR5), the projected rainfall data in late 20th (1979-2003) and 21st centuries (2075-2099) are simulated from TCCIP, and the hourly rainfall at 5 km horizontal resolution is provided for the end user. However, this resolution is inadequate for the


simulation of landslides or debris flow. To achieve precision for further calculations, the spatial interpolation from 5 km to 40 m is made for the selected scenarios and used as inputs for landslide simulation.

## 2.2 Landslide Inventory Simulation

The potential landslide simulation model is used to evaluate the probability of a landslide. It encompasses two major

approaches: statistical and physical. The common statistical approach employs a regression model, such as binary regression, to identify a set of maximum likelihood parameters based on historical data to predict the landslide distribution (Chang and Chiang, 2009). In the physical approach, the infinite slope stability theory is applied to calculate the safety factor and predict the potential landslide area, such as the transient rainfall infiltration and grid-based regional slope-stability model (TRIGRS) (Baum et al., 2008) and digital terrain model for mapping the pattern of potential shallow slope instability (SHALSTAB)

(Montgomery and Dietrich, 1994). Because the rainfall input in each grid cell is nonhomogeneous over space and time, a grid-based model could be practically useful for establishing the connection between rainfall and landslides. In this study, a physical approach using the TRIGRS model was selected for assessing the landslide area.

The TRIGRS is an inventory of shallow landslide simulation programs developed by U.S. Geological Survey (Baum et al., 2008), and it has been widely used in several case studies of shallow landslides (Salciarini et al., 2006;Godt et al., 2008;Park

et al., 2013). The potential slope failure can be determined from the ratio of resisting basal Coulomb friction and gravitationally induced driving stress, based on the infinite slope stability hypothesis. The ratio is called the factor of safety (FS) and can be expressed as

$$FS(Z,t) = \frac{\tan\phi}{\tan\alpha} + \frac{C - \psi(Z,t)\gamma_w \tan\phi}{\gamma_s d_{LZ} \sin\alpha \cos\alpha} , \tag{1}$$

where $\phi$ is the soil internal friction; $C$ is the cohesion; $\gamma_w$ and $\gamma_s$ are the unit weights of water and soil; $d_{LZ}$ is the depth of

the impermeable lower boundary or soil thickness; $\alpha$ is the slope angle; and $\psi(Z,t)$ is the pore water pressure with soil thickness $Z$ at time $t$. In addition, $Z$ is in the direction of the vertical coordinate and equal to $z\cos\alpha$, where $z$ is in the direction of the slope-normal coordinate. To calculate the pore water pressure with vertical infiltration in Eq. (1), a linearized solution of one-dimensional Richard's equation (Baum et al., 2008) is used, as furnished below:

$$\psi(Z,t) = [Z-d]\beta$$
$$+2\sum_{n=1}^{N}\frac{I_{nZ}}{K_Z}H(t-t_n)[D_1(t-t_n)]^{\frac{1}{2}}\sum_{m=1}^{\infty}\left\{ierfc\left[\frac{(2m-1)d_{LZ}-(d_{LZ}-Z)}{2[D_1(t-t_n)]^{1/2}}\right] + ierfc\left[\frac{(2m-1)d_{LZ}+(d_{LZ}-Z)}{2[D_1(t-t_n)]^{1/2}}\right]\right\} ,$$
$$-2\sum_{n=1}^{N}\frac{I_{nZ}}{K_Z}H(t-t_{n+1})[D_1(t-t_{n+1})]^{\frac{1}{2}}\sum_{m=1}^{\infty}\left\{ierfc\left[\frac{(2m-1)d_{LZ}-(d_{LZ}-Z)}{2[D_1(t-t_{n+1})]^{1/2}}\right] + ierfc\left[\frac{(2m-1)d_{LZ}+(d_{LZ}-Z)}{2[D_1(t-t_{n+1})]^{1/2}}\right]\right\} \tag{2}$$

where $d$ is the depth of the steady-state water table measured in the $Z$ direction; $\beta = \cos^2\alpha - (I_{ZLT}/K_S)$; $K_S$ is the saturated hydraulic conductivity in the $Z$ direction; $I_{ZLT}$ is the initial surface flux; $I_{nZ}$ is the surface flux of a given intensity for the $n^{th}$ time interval (rainfall input); $D_1 = D_0\cos^2\alpha$, where $D_0$ is the saturated hydraulic diffusivity; $N$ is the total





number of time intervals; and $H(t - t_n)$ is the Heaviside step function. The $ierfc$ is the first integral of complementary error function and can be expressed as

$$ierfc(\eta) = \pi^{-1/2} \exp(-\eta^2) - \eta erfc(\eta).$$

During the TRIGRS simulation, the FS of each grid is larger than one; specifically, the infinite slope is stable in the beginning. With the onset of rainfall and infiltration, the FS decreases because of an increase in the pore water pressure. Grid instability or infinite slope failure occurs when the FS is less than one, and it is regarded as a potential landslide area. The potential landslide volume or mass of the initial debris flow could be further evaluated for the debris flow simulation using the soil thickness $d_{LZ}$ in each unstable grid.

## 2.3 Debris Flow Simulation

For debris flow assessment, the empirical formula (Ikeya, 1981) has been resorted to. However, in recent years, because of the progress made in computer technology, numerical simulation has turned out to be a powerful approach. Various kinds of useful numerical programs have been developed by research and academic institutions worldwide. Although they are based on different theories, the governing equations are derived from mass and momentum conservations (Hutter and Greve, 2017;O'Brien et al., 1993;Hungr, 1995;Sassa et al., 2004;Liu and Huang, 2006;Nakatani et al., 2008;Armanini et al., 2009;Christen et al., 2010). According to the practical application of inputs, the models can be divided into hydrological- and geologic-based models. In hydrological-based models, the debris flow is simulated with a calibrated hydrograph at a specified inflow location, and it is particularly applied for channelized debris flow or mud flow. However, in the geologic-based models, debris flow starts from an initial mass distributed in its source area, such as landslide-triggered debris flow or an avalanche. To link the debris flow simulation with the landslide results, a geologic-based model is more useful. Hence, in this study, we applied the Debris-2D numerical simulation program (Liu and Huang 2006) for simulating the debris flow transport process. Debris-2D was developed by Liu and Huang (2006) and has been widely applied by researchers (Liu et al., 2009;Tsai et al., 2011;Wu et al., 2013). The debris flow is treated as a single phase non-Newtonian fluid in the model. A three-dimensional constitutive relationship introduced by Julien and Lan (1991) is adopted. With constitutive relationship and shallow water assumption, the governing equations for shear layer and plug layer can be derived from mass and momentum conservation. However, the shear layer thickness is usually less than 10% of the total flow depth (Liu and Huang, 2006). Therefore, the calculation of shear layer could be ignored and the depth-averaged governing equations of the leading order are written as follows:

$$\frac{\partial H}{\partial t} + \frac{\partial (uH)}{\partial x} + \frac{\partial (vH)}{\partial y} = 0, \tag{3}$$

$$\frac{\partial (uH)}{\partial t} + \frac{\partial (u^2 H)}{\partial x} + \frac{\partial (uvH)}{\partial y} = -gH \cos\theta \left( \frac{\partial B}{\partial x} + \frac{\partial H}{\partial x} \right) + gH \sin\theta - \frac{1}{\rho} \frac{\tau_0 u}{\sqrt{u^2 + v^2}}, \tag{4}$$





$$\frac{\partial(vH)}{\partial t} + \frac{\partial(uvH)}{\partial x} + \frac{\partial(v^2H)}{\partial y} = -gH\cos\theta\left(\frac{\partial B}{\partial y} + \frac{\partial H}{\partial y}\right) - \frac{1}{\rho}\frac{\tau_0 v}{\sqrt{u^2+v^2}},$$  (5)

where $H$ is flow depth; $B$ is the bed topography; $u$ and $v$ are the depth-averaged velocities in the $x$- and $y$-directions respectively; $\theta$ is the bottom bed slope; $\tau_0$ and $\rho$ are the stress and density of the debris-flow yield that are assumed to be constant; and $g$ is the gravitational acceleration. In these governing equations, the bottom erosion and deposition effects are

neglected. Upon omitting the shear layer, the yield stress becomes the dominant bottom stress. By applying equations (3)–(5), $H$, $u$, and $v$ can be calculated for an initial stationary debris pile $H$. In addition, debris flow obeys a condition (Liu and Huang, 2006) and starts to flow only when the pressure and gravitational effects exceed the yield stress effect.

For equations (3)–(5), the input data are topography $B$, initial debris flow depth $H$, and yield stress $\tau_0$. The topography is studied using the digital elevation model (DEM), which is the same as landslide simulation. Because extreme climates are

considered, enough rainfall is likely to initialize all the dry debris produced from the landslide. With the landslide area simulated by TRIGRS and its corresponding soil thickness $d_{LZ}$ in each grid cell, the initial debris flow depth $H$ can be determined from the following equation (Liu and Huang, 2006;Liu et al., 2009):

$$H = d_{LZ}/C_{d\infty},$$  (6)

where $C_{d\infty}$ is the equilibrium concentration (Takahashi, 1981), which is calculated as follows:

$$C_{d\infty} = \frac{\rho\tan\theta}{(\sigma-\rho)(\tan\phi-\tan\theta)}$$  (7)

where $\rho$ and $\sigma$ are the densities of water and sediment; $\phi$ is the internal friction angel; and $\theta$ is the average creek bottom slope.

**2.4 Loss assessment**

Conventionally, slope land losses are divided into direct and indirect losses. For assessing the direct loss from the debris flow

disasters in Taiwan's mountainous regions, a method was devised by Liu et al. (2009). The major losses that are assessed include construction, agriculture, forest, transportation, hydraulic, and tourism losses. Li and Yang (2010) built a household loss regression model for debris flow depending on actual survey data. This model considered several statistically significant variables, including the coverage area, height of debris flow coverage, number of people per household, and type of construction material (RC), for assessing the actual household loss incurred from the debris flow.

Based on experience from historical typhoon events, the major indirect impact of slope land disasters on Taipei is water shortages. This study emphasized the economic effects of water supply disruption. According to the questionnaire survey of Typhoon Soudelor (Li et al., 2016), 54.7% of all Taipei households (approximately one million) were affected by the disrupted water supply resulting from high turbidity. The average economic loss per household caused by water supply shortage is US$ 100. The majority of this expenditure was to buy clean drinking water; people also had to spend on water for routine cleaning





and washing. This study evaluated the water supply shortage (based on the survey data of Typhoon Soudelor), including the percentage of affected households in Taipei and the loss faced per household. All methods of quantifying losses are listed in Table 1, and the ensuing loss assessment of slope land disasters is based on this table.

5  **Table 1: Property Loss Function of Slope Land Disasters**

| Type | Formula for different use | Parameters | Value for assessment |
|---|---|---|---|
| **Direct Loss** | Household Use $$HL = \exp\left[\begin{array}{l} 9.36 + 0.736 \cdot \ln(LC) \\ +0.603 \cdot \ln(HC) \\ +0.21 \cdot DE + 0.092 \cdot NHP \\ -1.015 \cdot HT - 0.231 \cdot CP \\ +0.451 \cdot CT \end{array}\right]$$ | HL: household loss (NT dollar) <br> LC: landslide coverage (m²) <br> HC: height of coverage (m) <br> DE: disaster experience <br> NHP: number of household people <br> HT: house type <br> CP: community preparedness <br> CT: construction type | Household loss function (Li and Yang, 2010) |
| | Agriculture Use $$CL = \sum_{i=1}^{n} \alpha_i \left( CO_i \times CP_i \times CLA_i \right)$$ | CL: total cropper loss (NT dollar) <br> CO: cropper output for i$^{\text{th}}$ crop (Kg/ha) <br> CP: cropper price for i$^{\text{th}}$ crop (NT dollar/kg) <br> CLA: loss area for i$^{\text{th}}$ crop (ha) <br> $\alpha_i$ : modify coefficient <br> $i$ : the index for different crops within hazard area | Agriculture product price from Government (Liu et al., 2009) |
| | Forest Use $$FL = \sum_{i=1}^{n} \alpha_i \left( FAL_i \times DLA_i \right)$$ | FL: total foresty loss (NT dollar) <br> FAL: forestry loss from previous year (NT dollar/ha) <br> DLA: disaster area (ha) <br> $\alpha_i$ : modify coefficient <br> $i$ : different type of trees | Forest loss from the Forestry Bureau (Liu et al., 2009) |
| | Industry and Commerce Use $$ICL = \sum_{i=1}^{n} \alpha_i \left( ICP_i \times ICLA_i \right)$$ | ICL: industry and commerce loss (NT dollar) <br> ICP: industry and commerce price (NT dollar/m²) <br> ICLA: industry and commerce loss area (m²) <br> $\alpha_i$ : modify coefficient <br> $i$ : different place (county) | Industry and commerce product price from Government |
| | Building Use (Public) $$BL = \sum_{i=1}^{n} \sum_{j=1}^{m} \alpha_i \left( BC_{ij} \times BLA_{ij} \right)$$ | BL: building loss (NT dollar) <br> BC: building cost (NT dollar/m²) <br> BLA: building loss area ( m²) <br> $\alpha_i$ : modify coefficient <br> $i$ : different place (county) <br> $j$ : different building | Government's bulletin (Liu et al., 2009) |
| | Transportation and Hydraulic Use $$THL = \sum_{i=1}^{n} \alpha_i \left( SUC_i \times SLN_i \right)$$ | THL: traffic and hydraulic loss (NT dollar) <br> SUC: structure unit cost (NT dollar/m or NT dollar/m²) <br> SLN: structure loss number(m or m²) <br> $\alpha_i$: modify coefficient <br> $i$ : different structure | Transportation and Hydraulic loss function (Liu et al., 2009) |
| **Indirect Loss** | Water Supply Shortage $$WSSL = \sum_{i=1}^{n} \alpha_i \left( SD_i \times CUD_i \right)$$ | WSSL: water supply shortage loss (NT dollar) <br> SD: shortage day (Day) <br> CUD: consumption per unit day (NT dollar/day) <br> $\alpha_i$: modify coefficient <br> $i$ : different households | Survey data (Li et al., 2016) |



## 2.5 Integrated simulation process

An integrated simulation process was constructed as depicted in Fig. 1. The rainfall events were simulated using MRI-AGCM 3.2, downscaled with WRF, and modified by bias correction. To accomplish the study's objectives regarding climate change and its impacts, the simplest method was to study two extreme rainfall scenarios from different periods and compare them. This scenario approach was adopted for integrated simulation.

With the aid of rainfall and other relevant parameters, the potential landslide area was simulated using TRIGRS and the landslide inventory maps were constructed for various scenarios. Under extreme rainfall and loose landslide mass, sediment transport by debris flow from upstream catchment was assumed. Accordingly, all landslide masses were considered as input debris flow and simulated using Debris-2D. Finally, based on the above results, the economic losses incurred from the slope land disaster were evaluated, including both direct and indirect losses.

The integrated simulation in Fig. 1 provided a comprehensive view of the chain reaction. The simulation results from the current and future scenarios were compared in terms of climate change, and the compounded disasters were calculated for the extreme climatic events.

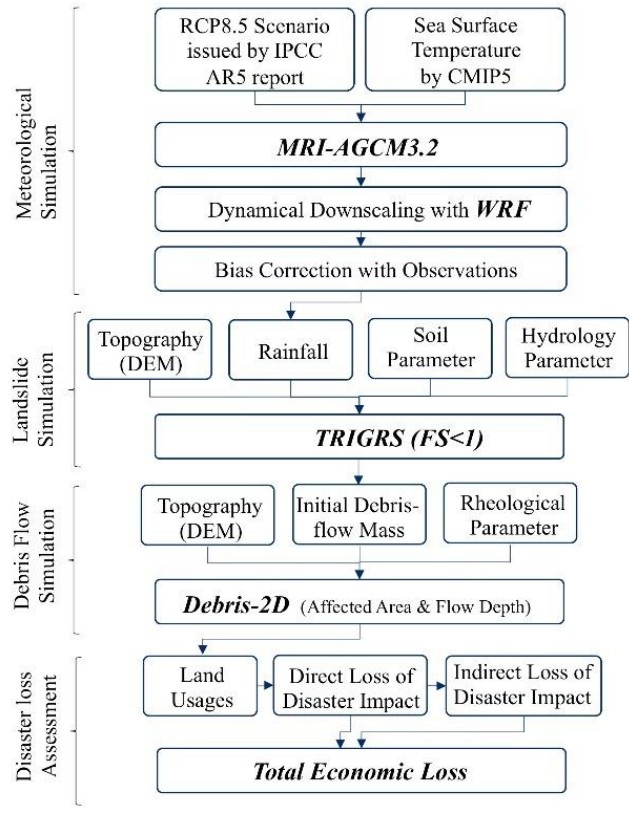

**Figure 1: Integrated Simulation Process**





## 3 Case Study: Xindian Watershed

### 3.1 Study Area

Xindian watershed is located upstream of Taipei City in Northern Taiwan. The river is one of the three major tributaries of the Tamsui River, and it is also the main source of drinking water for Taipei and New Taipei cities. According to the Taipei City

Running Water Center, over one million Taipei households rely on this river for their drinking water requirements. The chief tributaries of the Xindian River are Nanshi and Beishi, as represented in Fig. 2. Comparing these two tributaries, the Nanshi River catchment area is more fractured than the Beishi River catchment, and historically, landslides were rampant along the Nanshi River (Fig. 3). Hence, in this study, we focused on the Nanshi River catchment and ignored the areas beyond the Feitsui reservoir.

The study region depicted in Fig. 2 spans an area of 49,000 ha. Villages such as Wulai, Xinxian, and Fushan are located along the Nanshi River with 2716, 622, and 739 inhabitants, respectively. In this area, the elevation varies considerably and canyon-like topography can be noticed along the banks of the river. The study area is mainly located in the Tatungshan (Tt), Szeleng Sandstone (Em), Kangkou (Kk), Mushan (Ms), and Tsuku formations (Tu). Its contents include sandstone, argillite, slate, shale, and siltstone, as represented in Fig. 4. The age of the geological setting is between the Holocene and Eocene epochs. Numerous

folds and faults occur in this region. Because of the soft and fractured geological conditions prevailing in the Nanshi River catchment, geodisasters and the resultant sedimentation are major concerns in the area.

Considering the abovementioned facts, the Xindian watershed is an appropriate area for studying the impacts of slope land hazards. The economic impacts of sediment-related hazards are not only restricted to this area but also felt in the downstream cities, particularly in Taiwan's capital. For long-term city planning, it is of utmost importance to comprehend the whole

situation and devise suitable strategies, after due consideration of climate change aspects. Accordingly, the integrated simulation built in the previous section was applied for analysing the area. The model calibration and simulation results are presented in the following sections.



**Figure 2: Topography (shaded relief) and Location of the Study Area**



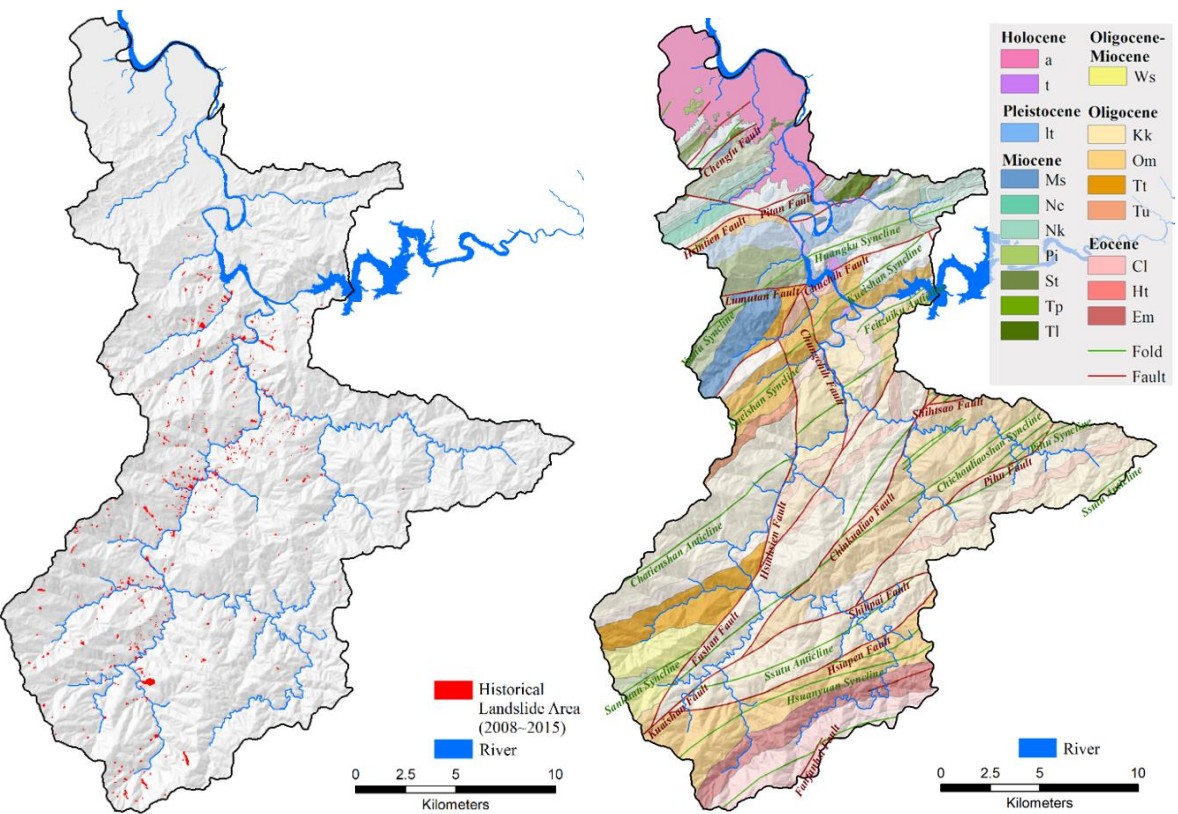

**Figure 3: Historical Landslide Area from 2008 to 2015 (Source: Central Geological Survey, Taiwan)**     **Figure 4: Lithological Map - 1:50,000 Scale**

## 3.2 Extreme Rainfall Scenarios

The rainfall projections for the late 20th (1979–2003) and 21st centuries (2075–2099) collected from TCCIP were chosen for comparison in our study area. Because bias correction was accomplished in a relevant study (Su et al., 2016), the data were directly used for our application. The top 20 rainfall events of the late 20th and 21st centuries are presented in Fig. 5. A pattern of increasing rainfall can be observed in the top five rainfall events.

To explore the potential impacts of the slope land problem in extreme climatic conditions, the worst cases (rank one rainfall event) from the late 20th and 21st centuries were selected for comparison. They are referred to as scenario 1 and scenario 2, respectively, in the following discussion. The distribution of the accumulated rainfall for both scenarios is shown in Fig. 6. Based on data provided by the Fushan meteorological station, the maximum accumulated rainfall was 911.4 mm in 61 hours and 1531.1 mm in 40 hours for scenario 1 and scenario 2, respectively. Similarly, the maximum intensities were 49.7 and 125.8 mm/hr for scenario 1 and scenario 2, respectively. For both scenarios, the temporal and spatial resolutions were in 1 hr and 5 km, respectively. Spatial interpolation from 5 km to 40 m was adopted for the purpose of slope land simulation.





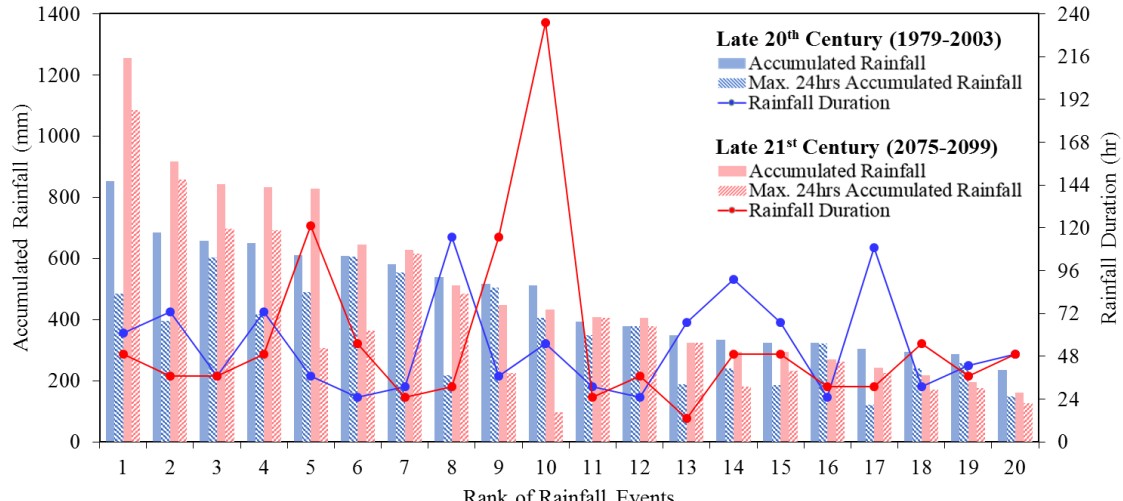

**Figure 5: Top 20 Rainfall Events of the Late 20th and 21st Centuries**

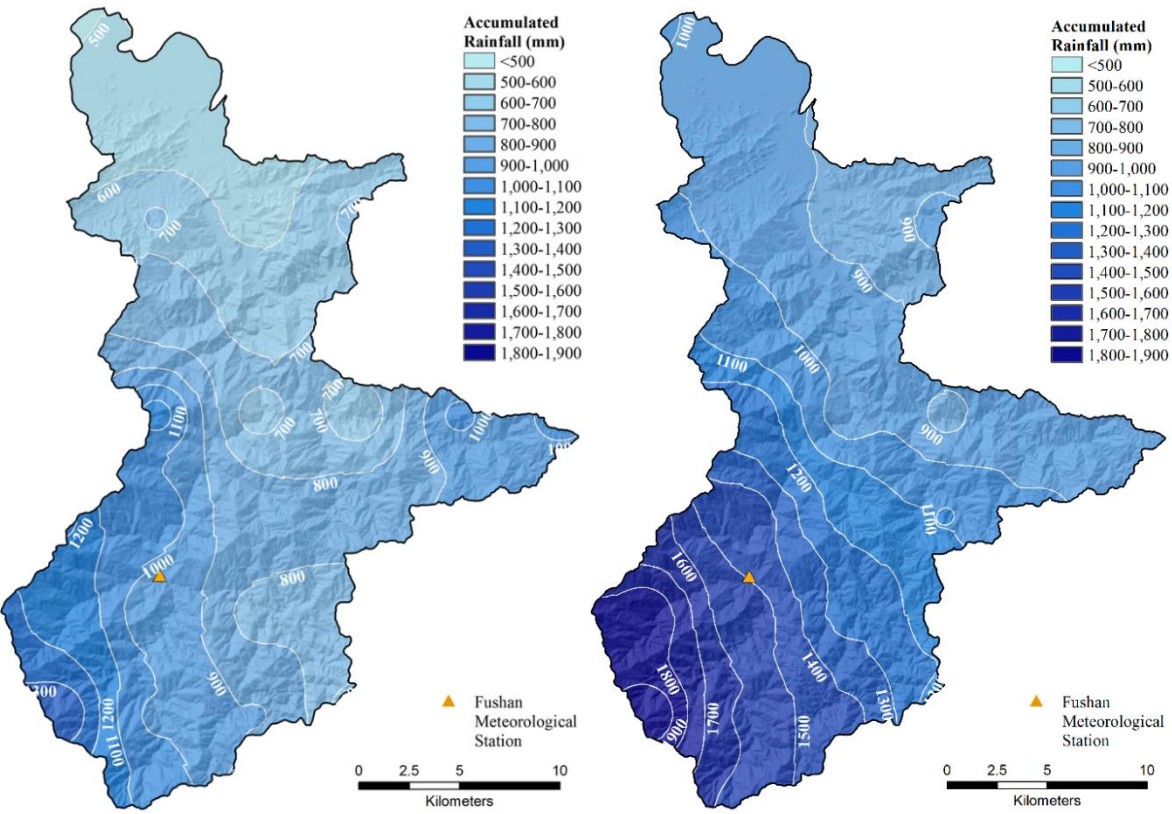

**Figure 6: Accumulated Rainfall Distribution (Left: Rank 1 rainfall of late 20th century; Right: Rank 1 rainfall of late 21st century)**



### 3.3 Landslide Inventory Simulation

Based on equations (1) and (2), the TRIGRS input data of each grid cell was separated into four parts: rainfall intensity, topographic information, soil, and hydraulic parameters. The rainfall intensity $I_{nZ}$ (mm/hr) presented in the previous section was directly used. The topographic information, such as slope $\alpha$ and flow aspect, were derived from DEM at a 40 m resolution.

The soil parameters of cohesion $C$, friction angle $\phi$, and unit weight $\gamma_s$ were calibrated based on past events; the soil thickness $d_{LZ}$ was calculated using the empirical slope-depth relationship (Khazai and Sitar, 2000;National Science and Technology Center for Disaster Reduction, 2011). The hydraulic parameters of saturated conductivity $K_S$ and diffusivity $D_0$ for the various geologic conditions were cited from past investigations (Central Geological Survey, 2010). Without considering antecedent precipitation, the initial depth of the steady-state water table $d$ was assumed to be the same as that of the soil

thickness $d_{LZ}$, and the initial infiltration rate for soil was taken to be $10^{-8}$ (m/s) (Chen et al., 2005).

Because the parameters of $C$, $\phi$, and $\gamma_s$, are subject to geological changes, the study area was separated into 18 zones in the geologic map (1/50,000). Moreover, the ratio of historical landslides in each slope-unit was divided into five, to subdivide the calibration zones from 18 to 90. In this classification system, some of the geology is in the same class as the landslide rate; hence, the calibration zones were simplified from 90 to 56. For the defined zones, the modified success rate (MSR) provided

by Huang and Kao (2006) was used for calibration and validation, as shown below:

$$MSR(\%) = \frac{1}{2}\frac{N_1}{N_1 + N_2} + \frac{1}{2}\frac{N_4}{N_3 + N_4},\tag{8}$$

where $N_1$ and $N_2$ denote the areas of $FS < 1$ and $FS > 1$, respectively, for the historical landslide areas. Similarly, $N_3$ and $N_4$ represent the areas of $FS < 1$ and $FS > 1$, respectively, for the historical non-landslide areas. The success rates of landslides and non-landslides can be obtained using Eq. (8). The units $N_1 - N_4$ were calculated from the slope-units. This

prediction is considered successful if the MSR > 70%. With the objective function of MSR, the parameters in each zone could be optimized for the rainfall events provided by the Central Weather Bureau and the historical landslide data provided by the Central Geological Survey. However, the historical landslide data are only updated annually; hence, the sum of representative rainfall events in each year was used for calibration. The landslide of Typhoon Soudelor in 2015 was successfully validated with an MSR of 91%; the calibration and validation results are presented in Table 2. Therefore, the predictions of the landslide

model for the study area were accurate.




**Table 2. Calibration and Verification of the TRIGRS Model based on MSR Calculation**

|  | Year | Representative rainfall events in each year | MSR |
|---|---|---|---|
|  | 2008 | Typhoon Kalmaegi, Typhoon Jangmi, Typhoon Sinlaku | 88% |
|  | 2009 | Typhoon Parma, Typhoon Morakot | 87% |
| Events for Calibration | 2010 | Typhoon Megi, Typhoon Fanapi | 84% |
|  | 2011 | Typhoon Nanmadol, 1001 Rainfall | 84% |
|  | 2012 | Typhoon Saola | 86% |
| Events for Verification | 2015 | Typhoon Soudelor | 91% |

In the simulation of these two scenarios, an increasing trend was observed in the potential landslide areas, in terms of the accumulated rainfall with a time delay. Moreover, the rate of increase in the accumulated landslide ratio in scenario 2 was found to be higher than in scenario 1, which could be attributed to the effect of rainfall intensity, as illustrated in Fig. 7. The stable time periods with maximum accumulated landslide ratios for scenario 1 and scenario 2 were 65 and 40 hours, respectively. Based on the landslide simulation results (FS<1) and soil thickness $d_{LZ}$ in each grid, the landslide inventory maps were drawn, as depicted in Fig. 8, and used for debris flow simulation.

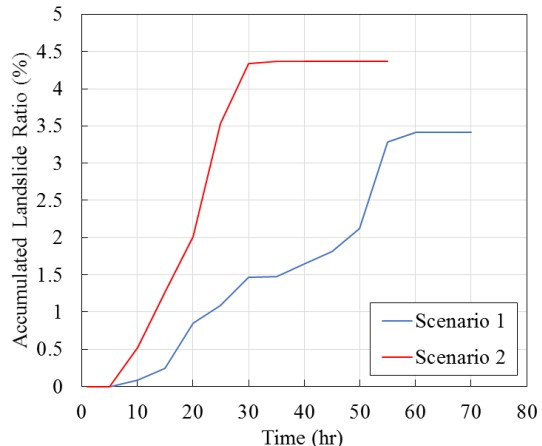

**Figure 7: Accumulated Landslide Ratio vs. Time for Scenarios 1 and 2 (landslide rate was calculated using the area of FS<1 over the whole area of the watershed)**





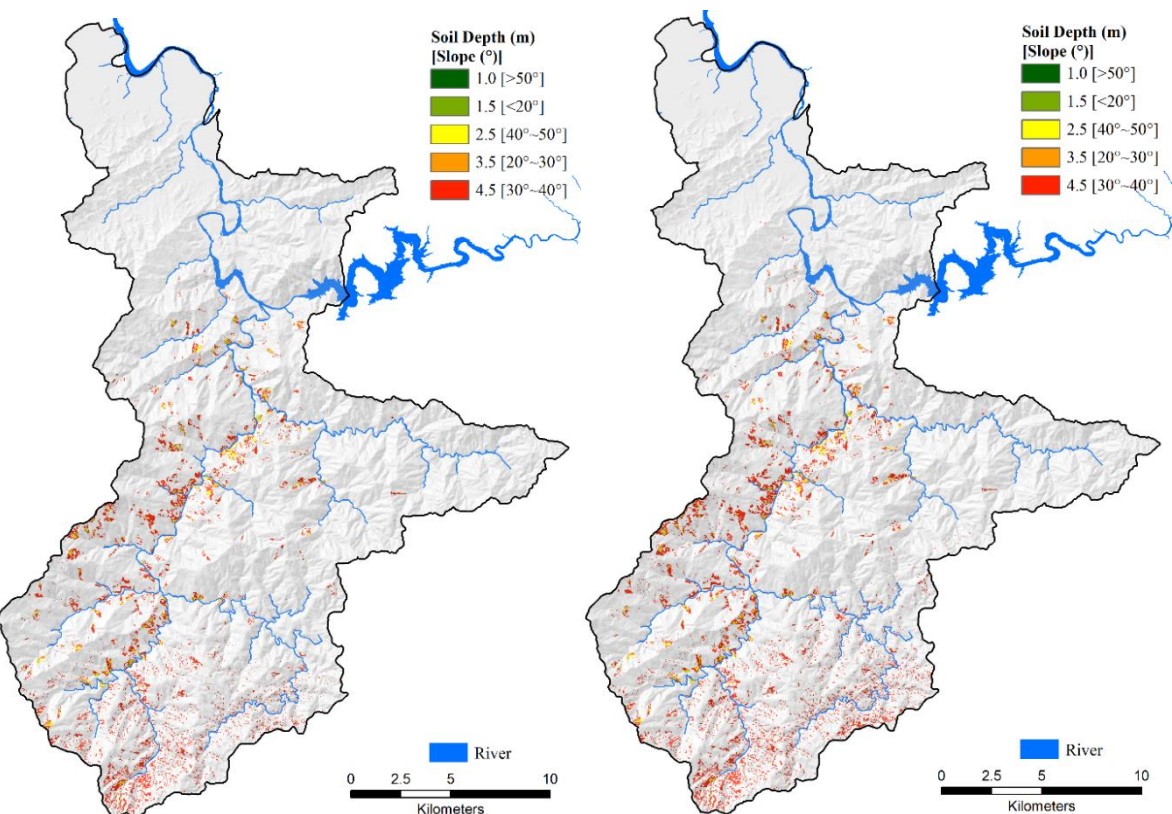

**Figure 8: Landslide Simulation Result by TRIGRS (left: scenario 1; right: scenario 2)**

## 3.4 Debris Flow Simulation

In Debris-2D simulations, the input data could be separated into three parts: topography (DEM), initial debris flow mass, and
rheological parameters. For calculating the initial debris flow mass, landslide inventory maps as well as Eqs (6) and (7) were
used. Because debris flow is triggered by extreme rainfall and loose landslide mass, the entire landslide mass was assumed to
be transformed into initial debris flow mass. Moreover, the concentration was presumed to be high and a maxima of
$C_{d\infty} \leq 0.603$ (Liu and Huang, 2006;Liu et al., 2009) was used for the practical estimations in the equations. However, the
occurrence of debris flow after slope failure cannot be predicted. Hence, the beginning of debris flow was assumed to be the
same as the starting time. The rheological parameter yield stress was estimated to be 800 Pa (Geotechnical Engineering Office,
2011) and used in the subsequent simulations.

The depth of debris flow in both scenarios is presented in Fig. 9 and its characteristics are described as follows. At a simulation
time of 5 min, the flow depth was over 20 m, and it occurred in the Zhakong River and in the midstream of the Nanshi River.
The source of the debris flow in the Zhakong River was an upstream landslide. However, the debris flow that occurred in the
midstream of the Nanshi River originated from the left bank landslide along the same river. At 10 min, all landslide debris was
flowing into the nearby tributary. In scenario 2, the front of the Zhakong River debris flow reached the tail end of the Nanshi



River debris flow at 15 min. After 20 min, the downstream Hapen River debris flow converged toward the Zhakong River debris flow. All debris flows started to decelerate and stopped completely at 30 min.

In both scenarios, the upstream Hapen River debris flows could not be transported downstream because of the meandering creek. The landslide debris along the downstream Hapen and Daluolan Rivers were mainly deposited at the junction of the

5 Zhakong, Daluolan, and Happen Rivers. The depths of the Daluolan and Happen Rivers made them insusceptible to the debris flow from the Zhakong or Nanshi Rivers. The front of the Zhakong River debris flow was deposited ahead of a sharp turn upstream of the Nanshi River in scenario 1, but it managed to reach the tail end of the Nanshi River debris flow in scenario 2. The Nanshi River debris flows were deposited at the junction of the Nanshi and Tonghou Rivers.

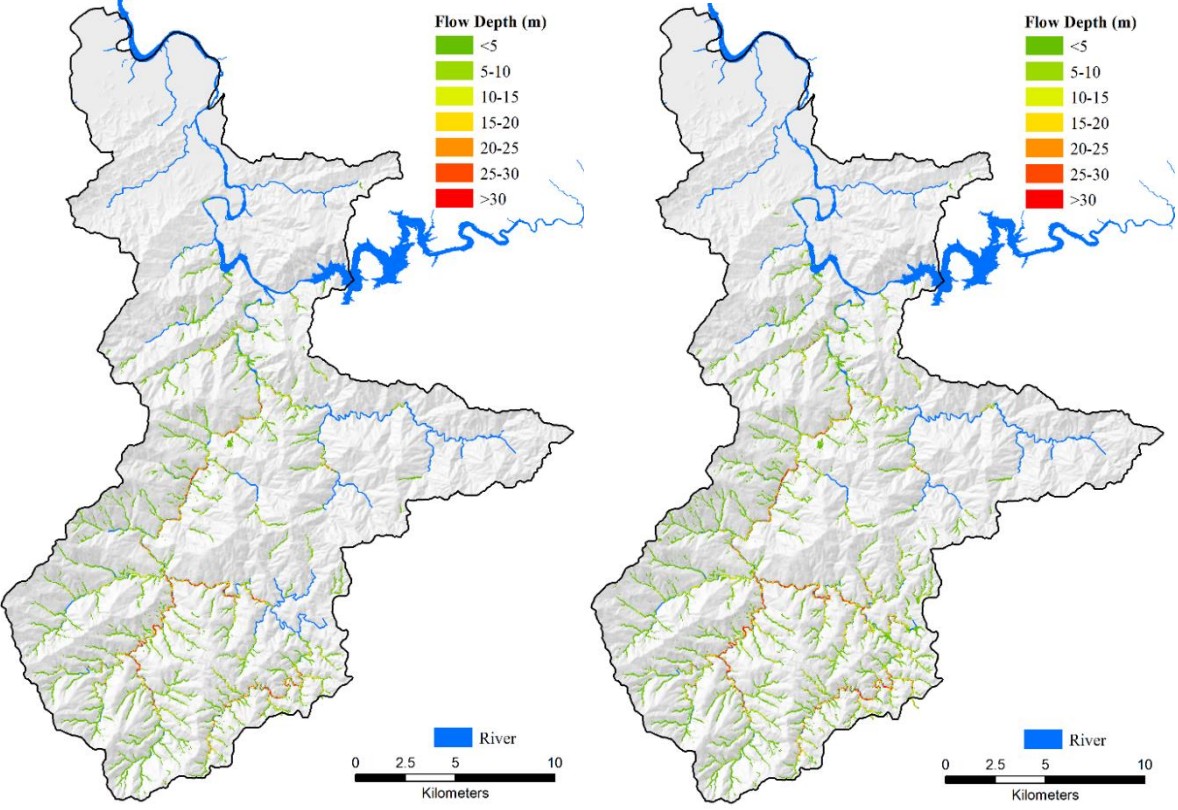

**Figure 9. Final Flow Depth of Debris Flow Simulated by DEBRIS-2D (left: scenario 1; right: scenario 2)**

## 4 Results and Discussion

### 4.1 Potential Effects on Natural Hazards

In scenarios 1 and 2, the grid-averaged maximum accumulated rainfalls were 853 mm in 61 hours and 1255 mm in 49 hours, respectively. Comparing the two scenarios, the grid-averaged accumulated rainfall was observed to increase by 402 mm, but

the duration decreased by 12 hours. Based on scenario 1, the decrement in duration and increment in grid-averaged




accumulated rainfall between the two scenarios were 19.67% and 47.22%, respectively. Hence, the characteristic of the rainfall changed to a shorter duration and greater intensity.

For the two rainfall scenarios, the total volume, including landslide and debris flow volumes, calculated from Eq. (6) were $1.18 \times 10^8$ m$^3$ and $1.52 \times 10^8$ m$^3$, respectively. The incremental volume between the two scenarios was 28.81%.

**Table 3. Comparison of Rainfall and Debris Flow Volume**

|  | Scenario 1 | Scenario 2 | Difference | Difference (%) |
|---|---|---|---|---|
| Rainfall Duration | 61hr | 49hr | -12hr | -19.67% |
| Accumulated Rainfall | 853mm | 1255mm | +403mm | +47.22% |
| Debris Flow Volume | $1.18 \times 10^8$ m$^3$ | $1.52 \times 10^8$ m$^3$ | $3.4 \times 10^7$ m$^3$ | +28.81% |

**4.2 Loss Assessment for Compounded Disasters**

Based on the simulated results in Table 3, possible economic losses were evaluated according to the quantified method of disaster loss in Table 1. The direct and indirect losses are illustrated in Fig. 10 and Fig. 11, respectively. According to Fig. 10, the main directs losses were to transportation, households, and public facilities. Because most of roads were built along the riverside, they were easily damaged by the landslide and debris flow. This transportation loss constituted the biggest impact of the disaster, as witnessed in Fig. 10. The transportation losses in scenario 2 were even more severe than in scenario 1, amounting to 12.32% and approximately US$ 52.14 million. The second largest were household losses; affected households were predominantly located along the riverside. In this case too, the losses in scenario 2 were more severe than in scenario 1, amounting to approximately 8.01% and US$ 22.21 million. The third largest loss was faced by public facilities, including power supply, water supply, hospitals, and schools.

Among the indirect losses, this study mainly focused on water supply shortage. According to the household survey data of Typhoon Soudelor in 2016 (Li et al., 2016), 54.7% of Taipei households were affected by turbid water caused by the landslide that occurred upstream. The total landslide volume was estimated to be $9.8 \times 10^6$ m$^3$ (Wu et al., 2016a) and caused water supply shortages for 42 hours. Therefore, compared with the debris flow volume in Table 3, the volumes in scenarios 1 and 2 were 12 and 15.5 times greater than Typhoon Soudelor, respectively. Because the capacity of water treatment plants to remove turbidity is fixed, the required treatment time for turbid water is proportional to the debris flow volume. Based on the comparison of the landslide volume with the Typhoon Soudelor, the water supply shortage time periods in scenarios 1 and 2 were 506 and 651 hours, respectively. Therefore, the total economic losses in scenarios 1 and 2 were evaluated to be US$ 1211.11 million and US$ 1560.07 million, respectively, as depicted in Fig. 11.




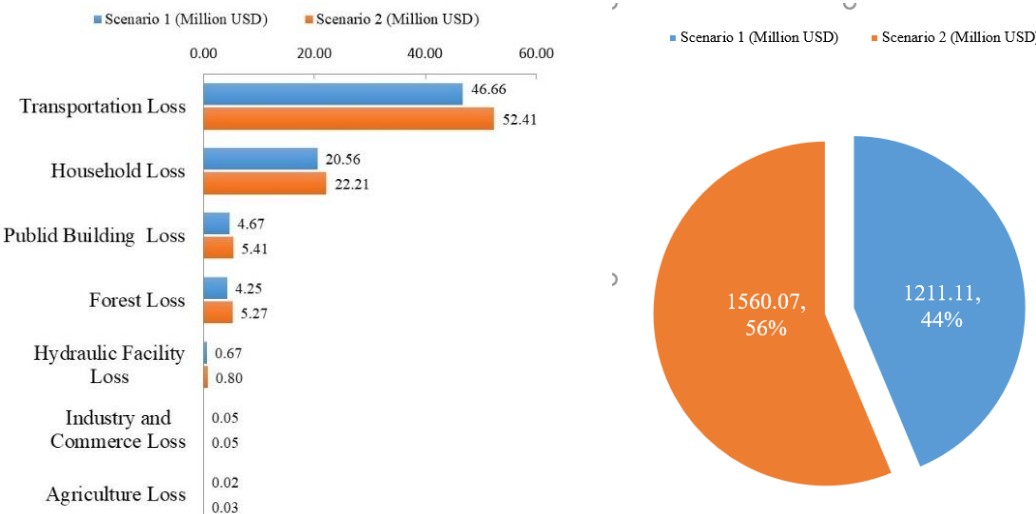

**Figure 10. Direct Losses**    **Figure 11. Indirect Losses**

Table 4 lists the total losses (direct and indirect) incurred in scenarios 1 and 2. Accordingly, the total loss faced in scenario 2 was approximately US$ 1646.25 million, which is greater than the US$ 1228.00 million loss faced in scenario 1. In other words, increased precipitation triggered by extreme events related to climate change is likely to cause more damage in the

5 future than at present, and the losses are projected to increase by 27.8% or US$ 358.25 million.

Furthermore, another problem worthy of discussion is that the indirect losses in scenarios 1 and 2 are both far greater than the direct losses. This means that the indirect losses are more damaging than the direct losses, and the ratios (indirect loss divided by direct loss) for scenarios 1 and 2 were calculated to be 15.75 and 18.1, respectively. In addition, the results substantiated that more serious disasters result in a higher proportion of indirect losses. Therefore, when discussing climate-induced slope

10 land impacts faced by Taipei, considering only direct losses will result in grossly underestimating the actual magnitude of the impact.

**Table 4. Comparison of Economic Losses**

| Land Use Type | | Scenario 1 (Million USD) | Scenario 2 (Million USD) | Difference (Million USD) |
|---|---|---|---|---|
| Direct Loss | Transportation Loss | 46.66 | 52.41 | 5.75 |
| | Household Loss | 20.56 | 22.21 | 1.65 |
| | Public Building Loss | 4.67 | 5.41 | 0.74 |
| | Forest Loss | 4.25 | 5.27 | 1.02 |
| | Hydraulic Facility Loss | 0.67 | 0.80 | 0.13 |
| | Industry and Commerce Loss | 0.05 | 0.05 | 0.00 |
| | Agriculture Loss | 0.02 | 0.03 | 0.01 |
| Indirect Loss | Water Supply Shortage Loss | 1211.11 | 1560.07 | 348.96 |
| Total Economic Loss | | 1288.00 | 1646.25 | 358.25 |




# 5 Conclusion

In recent years, slope land problems associated with climate change have become crucial topics of discussion. With the aid of climate projected scenarios, an integrated physical simulation process was proposed for analysing the potential impacts from a watershed point of view. The Xindian watershed in Taiwan was selected for studying the effects of extreme rainfall events.

Landslides, debris flow, and related loss assessments were executed in a step-by-step manner.

The rainfall scenarios in late 20th (1979-2003) and 21st centuries (2075-2099) were simulated using MRI-AGCM 3.2, and dynamic downscaling with WRF was adopted for producing the hourly rainfall data in a 5 km horizontal resolution. The resulting data were further interpolated from 5 km to 40 m for the landslide simulation input. The potential landslide area was varied according to the changing rainfall and was simulated using the TRIGRS model that was calibrated and validated based

on past events. Later, the corresponding debris flow volume was determined using the equilibrium concentration, and the debris flow was simulated with the help of the Debris-2D model. With the aid of these simulation results, multiple loss functions were applied for the evaluation of direct and indirect economic losses.

Upon comparing the worst cases of rainfall in the late 20th and 21st centuries, the grid-averaged accumulated rainfall increased by 47.22%, but the duration decreased by 19.67%. Considering the increased rainfall intensity in scenario 2, the estimated

volume caused by the chain impacts of landslides and debris flow was increased by 28.81%. Because of the increasing impacts of slope land disasters caused by climate change, the economic losses are projected to increase by 27.8% or US$ 358.25 million. Among all of the losses, the main direct losses were to transportation, households, and public facilities in mountainous areas; the chief indirect loss was the water supply shortage in urban areas. Notably, the value of the indirect losses in both scenarios was much greater than that for the direct losses. This means that whenever heavy rainfall causes slope land disasters in areas

other than Taipei Metropolitan Area, its influence on Taipei will be mainly in the form of indirect impacts. This study solely discussed the problem of water supply shortage in residential areas. If industrial and commercial losses are also included, the economic impacts will be increased manifold. Therefore, methods to improve the resilience of water resources, as well as the development of alternative water sources (such as establishing cross-regional water transfer mechanisms and groundwater wells), will be crucial adaptation strategies for the Taipei Metropolitan Area in response to slope land disaster impacts resulting

from climate change.

**Acknowledgments**

This research was supported by the Ministry of Science and Technology of Taiwan under MOST 105-2625-M-865-001.

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
