# Peer review of "Potential Impact of Climate Change and Extreme Events on Slope Land Hazard - A Case Study of Xindian Watershed in Taiwan"

_Natural Hazards and Earth System Sciences, 2018_

## Referee Comment (RC1) · Anonymous Referee #1 · 6 Jul 2018

This manuscript proposed the scenario approach coupled with landslide simulation, debris flow simulation and loss assessment. The reviewer believes that the approach to link the landslide simulation with debris flow simulation as well as loss assessment was unique and interesting but the manuscript should be improved to be published.

1. In some parts of the manuscript, the sentences used by the authors are unclear. The authors are advised to rephrase these sentences in a better way or proof reading by English native speaker. One example is the sentence on Lines 9 to 13 of page 2.

2. In line 7 of page 2, the authors mentioned that "some potential effects of landslides have been investigated by studying the differences between current and future scenarios". However, any detailed explanation about the potential effects was not provided, so the reviewer cannot understand the explanation.

3. In line 14 – 15 of page 2, the authors explained that the accumulated rainfall over a period of 3, 6, 12 hours exceeded the 200 year record. However, the authors did not provide exact rainfall amounts. To understand the climate condition of Taipei area, total amount of rainfall accumulations for 3, 6, 12 hour periods should be provided.

4. In line 13 of page 4, "The TRIGRS is an inventory of shallow landslide simulation programs developed . . ..". But the TRIGRAS is not an inventory. Inventory means that the collection of past landslide features in a certain area for a certain period.

5. Does the subtitle "Landslide Inventory Simulation" mean that the location of past landslides (That is, landslide inventory) were simulated and matched by TRIGRS? However, it does not seem that the analysis results of TRIGRS are matched to the location of past landslides in this manuscript. In this case, the term "landslide inventory simulation" should be revised.

6. In Fig. 3, the authors provided historical landslide area. The reviewer recommends to provide more detailed information about how the inventory was constructed and the landslide locations were obtained.

7. In 3.3, the soil parameters such as cohesion, friction angle, unit weight, hydraulic conductivity, and diffusivity were used as input values in landslide analysis using TRIGRS. However, any values for the soil parameters were not provided. Since the input parameters in TRIGRS are important, the values of the input parameters should be provided. In addition, since the soil thickness is also an important parameter affecting the simulation results, the detailed procedure to evaluate the soil thickness using slope-depth relationship should be explained.

8. The explanations in line 11 – 14 of page 13 is not clear. The reviewer cannot understand why and how calibration zones were reduced from 90 to 56. Please rewrite

the paragraph clearly.

9. In line 7 – 8 of page 14, "Based on the landslide simulation results and soil thickness in each grid, the landslide inventory map were drawn, as depicted in Fig. 8, . . .". In reviewer's opinion, Fig. 8 is not landslide inventory but landslide analysis results. In addition, the author should provide clear explanation that the procedure that the authors performed and the results in Fig. 8 that the authors obtained. Since the most analysis results using TRIGRS show the distribution of factor of safety, the reviewer cannot understand the reason that Fig. 8 shows the soil depth as the results of analysis. Were the soil depths in Fig. 8 used in debris flow simulation? In the manuscript, any explanations were not provided.

10. In line 7 – 11 of page 15, the reason that the concentration was presumed to be high and a maxima was used for the practical estimations should be explained. In addition, the meaning of the sentence "the beginning of debris flow was assumed to be the same as the starting time." is not clear.

11. In 4.2, the authors provides the calculation results of the possible economic losses using the quantified method in Table 1. However, the authors did not provide any values used in the calculations and the calculation procedures that the authors performed. The detailed information should be provided.

---

## Author Comment (AC1) · 23 Jul 2018

Reviewer 1: Anonymous

General comments: This manuscript proposed the scenario approach coupled with landslide simulation, debris flow simulation and loss assessment. The reviewer believes that the approach to link the landslide simulation with debris flow simulation as well as loss assessment was unique and interesting but the manuscript should be improved to be published.

Authors response: We are grateful for the helpful specific comments. These comments

should substantially improve the manuscript. Please see our response to the specific suggestions below.

Reviewer 1: In line 7 of page 2, the authors mentioned that "some potential effects of landslides have been investigated by studying the differences between current and future scenarios". However, any detailed explanation about the potential effects was not provided, so the reviewer cannot understand the explanation.

Authors: We will provide and explain the potential effects observed from past researches.

Reviewer 1: In line 14 – 15 of page 2, the authors explained that the accumulated rainfall over a period of 3, 6, 12 hours exceeded the 200 year record. However, the authors did not provide exact rainfall amounts. To understand the climate condition of Taipei area, total amount of rainfall accumulations for 3, 6, 12 hour periods should be provided.

Authors: The maximum 3, 6, 12, and 72-hour rainfall during Typhoon Soudelor in Fushan meteorological station are 253mm, 442mm, 655mm, and 792mm respectively.

Reviewer 1: In line 13 of page 4, "The TRIGRS is an inventory of shallow landslide simulation programs developed : : :.". But the TRIGRAS is not an inventory. Inventory means that the collection of past landslide features in a certain area for a certain period.

Authors: Thank you for your correction. We will rephrase the misused word "inventory" as "susceptibility."

Reviewer 1: Does the subtitle "Landslide Inventory Simulation" mean that the location of past landslides (That is, landslide inventory) were simulated and matched by TRIGRS? However, it does not seem that the analysis results of TRIGRS are matched to the location of past landslides in this manuscript. In this case, the term "landslide inventory simulation" should be revised.

Authors: Yes, this is what we mean. We will rephrase "inventory" as "susceptibility" to

avoid misunderstanding.

Reviewer 1: In Fig. 3, the authors provided historical landslide area. The reviewer recommends to provide more detailed information about how the inventory was constructed and the landslide locations were obtained.

Authors: The historical landslide area (landslide inventory) were delineated by aerial photo by Central Geological Survey in Taiwan annually.

Reviewer 1: In 3.3, the soil parameters such as cohesion, friction angle, unit weight, hydraulic conductivity, and diffusivity were used as input values in landslide analysis using TRIGRS. However, any values for the soil parameters were not provided. Since the input parameters in TRIGRS are important, the values of the input parameters should be provided. In addition, since the soil thickness is also an important parameter affecting the simulation results, the detailed procedure to evaluate the soil thickness using slope-depth relationship should be explained.

Authors: The calibrated parameters of TRIGRS are provided in the supplementary document, as shown in Table S1. The relationship of slope and landslide depth is based on the survey data in Taiwan (Chen et al, 2010), as shown in Table S2. We will provide the references and descriptions in the next version.

Reviewer 1: The explanations in line 11 – 14 of page 13 is not clear. The reviewer cannot understand why and how calibration zones were reduced from 90 to 56. Please rewrite

Authors: There are 18 geologic settings and we set the landslide rate in 5 classes. Because zero landslide rate occurred in some geologic settings, only 56 zones were obtained for parameters calibration. We will rephrase our description.

Reviewer 1: In line 7 – 8 of page 14, "Based on the landslide simulation results and soil thickness in each grid, the landslide inventory map were drawn, as depicted in Fig. 8, : : :". In reviewer's opinion, Fig. 8 is not landslide inventory but landslide analysis

results. In addition, the author should provide clear explanation that the procedure that the authors performed and the results in Fig. 8 that the authors obtained. Since the most analysis results using TRIGRS show the distribution of factor of safety, the reviewer cannot understand the reason that Fig. 8 shows the soil depth as the results of analysis. Were the soil depths in Fig. 8 used in debris flow simulation? In the manuscript, any explanations were not provided.

Authors: Yes, the Fig. 8 is the TRIGRS simulation results. The soil thickness is one of the inputs for TRIGRS, and it was used for calculating the initial debris-flow volumes in Debris-2D input. So we drew the TRIGRS simulation results with soil thickness in Fig. 8. We will add the description in the next version.

Reviewer 1: In line 7 – 11 of page 15, the reason that the concentration was presumed to be high and a maxima was used for the practical estimations should be explained. In addition, the meaning of the sentence "the beginning of debris flow was assumed to be the same as the starting time." is not clear.

Authors: We will add some description to explain the calculation procedure.

1. The equilibrium concentration of debris flow can be estimated by an empirical formula purposed by Takahashi (1981) in Eq. (7) and the maximum value cannot exceed 0.603 (Liu & Huang, 2003) which is occurred when the slope larger than 20.6âĹŸ. Due to the slope of our most study basin even more than 20.6âĹŸ, thus this paper direct take 0.603 for the concentration value of debris flow to estimate debris-flow volumes in Eq. (6).

2. In reality, the landslide triggered debris flows in different locations could be occurred in different time. However, it is difficult to predict the debris flow occurred time after landslide, and in this paper what we concern is the final volume and influence area caused by debris flows. Therefore, the assumption of all the debris flows be the same starting time would not affect the final results.

Reviewer 1: In 4.2, the authors provides the calculation results of the possible economic losses using the quantified method in Table 1. However, the authors did not provide any values used in the calculations and the calculation procedures that the authors performed. The detailed information should be provided.

Authors: The calculation procedures of economic losses could be divided into three parts. First, we need to identify the impact area and depth of disaster which were got from the simulation results of debris flow. Then, the debris flow coverage area will be intersected with land-use map for identifying the loss of different use (e.g. household use, agriculture use, forest use etc.). Finally, the losses could be evaluated by loss functions and the corresponding parameters established in the database according to the uses. The total losses is the summation of the individual losses in different uses. The debris flow coverage area for different land use are provided in the supplementary document, as shown in Table S3.

Please also note the supplement to this comment:
https://www.nat-hazards-earth-syst-sci-discuss.net/nhess-2018-125/nhess-2018-125-AC1-supplement.pdf
* * *

---

## Referee Comment (RC2) · Anonymous Referee #2 · 14 Aug 2018

This manuscript describes an integrated approach to forecast the the economic impact of shallow landslides and debris flows in the framework of climate change scenarios. I personally think that the approach proposed is very interesting and useful since provides in quantitative terms the loss related to landslides and debris flows based on well-established literature methods. Anyway I think that the manuscript should be revised and improved before to be accepted for publication in the journal. In general the manuscript is well written but a revision of the manuscript structure and the clarification of some weak points would make the manuscript more clear and readable.

Here below my comments:

[Figure]

1. I suggest you to revise the Methodology section. I think there is no need to describe in detail (with equations) the TRIGRS and Debris-2D models. In this section I would suggest you to explain better why you have selected these methods among all the literature ones and then refer to the original papers for further information about the model equations. Furthermore, at the beginning of sections 2.1, 2.2, 2.3 and 2.4 you provide a description of the state of the art. This parts should be moved in the Introduction. In general I suggest to shorten the methodology description, moving the state of the art in the Introduction, avoiding the description of the models and the subdivision in sub chapters (2.1, 2.2 and so on)

2. In line1-2 of page 4 you state that the "spatial interpolation from 5 km to 40 m is made for the selected scenarios and used as inputs for landslide simulation." What do you mean for spatial interpolation? Please clarify and provide more information.

3. The sentence in line 13-14 of page 4 is not correct since TRIGRS is not an inventory of shallow landslides simulation but a physically-based model to forecast shallow landslides occurrence under rainfall events. Please rephrase

4. The reviewer suggests to revise the term landslide in the methodology section. The landslides simulated by the TRIGRS model are shallow landslides. I think you should use this term instead of the general term landslide which include all types of landslides.

5. In Fig. 3 historical landslide area from 2008 and 2015 are reported. Please provide more information about how the inventory has been realized.

6. In line 4 of page 5 I suggest you to replace the term during with at the beginning.

7. In section 3.3 you don't provide any detailed information about the soil parameters used in the simulation of TRIGRS. In general in physically-based models the selection of soil parameters is an important issue. I suggest you to provide a table with soil parameters values and to describe how you have measured these data or which is the source.

8. The sentence in line 11-14 at page 13 is not clear, please rephrase.

9. In my opinion the title of Figure 8 is uncorrect since the TRIGRS model provides factor of safety maps and not a map of soil depth. Please provide a figure with the results of the simulation and specify better what the figure 8 represents.

10. The results of loss assessment provided in section 4.2 are very interesting, anyway a clear explanation on how they have been obtained is missing. Please clarify better this point, providing clear description of calculation procedure.
* * *

---

## Author Comment (AC2) · 22 Aug 2018

Reviewer 2: Anonymous

General comments:

This manuscript describes an integrated approach to forecast the the economic impact of shallow landslides and debris flows in the framework of climate change scenarios. I personally think that the approach proposed is very interesting and useful since provides in quantitative terms the loss related to landslides and debris flows based on well-established literature methods. Anyway I think that the manuscript should be revised and improved before to be accepted for publication in the journal. In general the manuscript is well written but a revision of the manuscript structure and the clarification of some weak points would make the manuscript more clear and readable.

Authors: We are grateful for the helpful specific comments. These comments should substantially improve the manuscript. Please see our response to the specific suggestions below.

Specific comments:

Reviewer 2: I suggest you to revise the Methodology section. I think there is no need to describe in detail (with equations) the TRIGRS and Debris-2D models. In this section I would suggest you to explain better why you have selected these methods among all the literature ones and then refer to the original papers for further information about the model equations. Furthermore, at the beginning of sections 2.1, 2.2, 2.3 and 2.4 you provide a description of the state of the art. This parts should be moved in the Introduction. In general I suggest to shorten the methodology description, moving the state of the art in the Introduction, avoiding the description of the models and the subdivision in sub chapters (2.1, 2.2 and so on)

Authors: Thank you for your suggestion. We will revise the introduction section and methodology section in the next version.

Reviewer 2: In line1-2 of page 4 you state that the "spatial interpolation from 5 km to 40 m is made for the selected scenarios and used as inputs for landslide simulation." What do you mean for spatial interpolation? Please clarify and provide more information.

Authors: In TRIGRS simulation, the 40m*40m DEM was used as topography input. However, the spatial resolution of rainfall were in 5km*5km. To satisfy the spatial resolution as 40m*40m, the rainfall was interpolated by inverse distance weighting (IDW) method from 5km*5km to 40m*40m. We will rephrase the description in the next version.

Reviewer 2: The sentence in line 13-14 of page 4 is not correct since TRIGRS is not an inventory of shallow landslides simulation but a physically-based model to forecast shallow landslides occurrence under rainfall events. Please rephrase

Authors: Thank you for your correction. We will rephrase the sentence according to the suggestion.

Reviewer 2: The reviewer suggests to revise the term landslide in the methodology section. The landslides simulated by the TRIGRS model are shallow landslides. I think you should use this term instead of the general term landslide which include all types of landslides.

Authors: Thank you for your suggestion. We will follow your suggestion to revise.

Reviewer 2: In Fig. 3 historical landslide area from 2008 and 2015 are reported. Please provide more information about how the inventory has been realized.

Authors: The historical landslide area (landslide inventory) were delineated by aerial photo by Central Geological Survey in Taiwan annually. We will add description in the next version.

Reviewer 2: In line 4 of page 5 I suggest you to replace the term during with at the beginning.

Authors: Thank you for your suggestion. We will rephrase this sentence in the next version.

Reviewer 2: In section 3.3 you don't provide any detailed information about the soil parameters used in the simulation of TRIGRS. In general in physically-based models the selection of soil parameters is an important issue. I suggest you to provide a table with soil parameters values and to describe how you have measured these data or which is the source.

Authors: Yes, the parameters are very important. The calibrated parameters of TRI-

GRS are provided in the supplementary document, as shown in Table S1.

Reviewer 2: The sentence in line 11-14 at page 13 is not clear, please rephrase.

Authors: There are 18 geologic settings and we set the landslide rate in 5 classes, therefore totally we have 90 zones. However, there are some geologic settings are stable without landslides. So the total zones decrease to 56 zones for parameter calibration. We will rephrase our description.

Reviewer 2: In my opinion the title of Figure 8 is uncorrect since the TRIGRS model provides factor of safety maps and not a map of soil depth. Please provide a figure with the results of the simulation and specify better what the figure 8 represents.

Authors: In Fig. 8, the shallow landslide area were simulated by TRIGRS. The soil depth were provided as reference because it is one of the input parameter of TRIGRS and DEBRIS-2D. We will rephrase the title of Fig. 8 and give more description to avoid misunderstanding.

Reviewer 2: The results of loss assessment provided in section 4.2 are very interesting, anyway a clear explanation on how they have been obtained is missing. Please clarify better this point, providing clear description of calculation procedure.

Authors: The calculation procedures of economic losses could be divided into three parts. First, we need to identify the impact area and depth of disaster which were got from the simulation results of debris flow. Then, the debris flow coverage area will be intersected with land-use map for identifying the loss of different use (e.g. household use, agriculture use, forest use etc.). Finally, the losses could be evaluated by loss functions and the corresponding parameters established in the database according to the uses. The total losses is the summation of the individual losses in different uses. The debris flow coverage area for different land use are provided in the supplementary document, as shown in Table S3.

Please also note the supplement to this comment:
https://www.nat-hazards-earth-syst-sci-discuss.net/nhess-2018-125/nhess-2018-125-AC2-supplement.pdf

[Figure]

**Supplement:**

**Supplement of TRIGRS input parameters**

**Table S1 The parameters used in TRIGRS**

| Geologic Time | Name (abbr.) | $\gamma_s$ (kN/m³) | C (kPa) | $\phi$ (°) | K ($10^{-6}$ m/s) | D ($10^{-6}$ m²/s) | Description (Ref: Central Geological Survey in Taiwan) |
|---|---|---|---|---|---|---|---|
| Holocene | Alluvium (a) | 19.5 | 10.5 | 34 | 29 | 8800 | Gravel, sand, and mud |
| | Terrace Deposits (t) | 23 | 6.5 | 30 | 0.7 | 220 | Gravel, sand and clay |
| Pleistocene | Lateritic Terrace Deposits (lt) | 18.6 | 35 | 30 | 0.8 | 800 | Red earth, lateritic gravel, sand, intercalated with sand and silt lentils |
| Miocene | Mushan Formation (Ms) | 27.5 | 16.8-28.8 | 32.0-36.0 | 10 | 2000 | Alternations of sandstone and shale, intercalated with coal seams |
| | Nanchuang Formation (Nc) | 27.5 | 23.5 | 34.5 | 10 | 2000 | Alternations of sandstone and shale, intercalated with coal seams |
| | Nankang Formation (Nk) | 27.5 | 29 | 36 | 10 | 2000 | Sandstone, siltstone, and shale |
| | Piling Shale (Pi) | 24.8 | 19.9-27.4 | 32.0-35.0 | 10 | 2000 | Shale with intercalated sandstone |
| | Shihti Formation (St) | 27.5 | 24.1-30.1 | 32.0-34.0 | 10 | 2000 | Alternations of sandstone and shale, intercalated with coal seams |
| | Tapu Formation (Tp) | 27.5 | 20.9 | 34 | 10 | 2000 | Alternations of muddy sandstone, white sandstone and shale |
| | Taliao Formation (Tl) | 27.5 | 16.3-27.3 | 32.0-36.0 | 10 | 2000 | Shale and sandstone |
| Oligocene-Miocene | Wenshui Formation (Ws) | 24.8 | 16.4-28.9 | 32.0-36.0 | 10 | 2000 | Sandstone and shale interbeds |
| Oligocene | Kangkou Formation (Kk) | 25.3 | 20.6-33.1 | 26.0-31.5 | 20 | 4000 | Argillite or slate intercalated with thin to thick-bedded siltstone |
| | Shuichangliu Formation (Om) | 27.5 | 21.0-33.5 | 29.0-33.0 | 10 | 2000 | Argillite, slate |
| | Tatungshan Formation (Tt) | 27.5 | 19.1-33.0 | 28.0-34.0 | 10 | 2000 | Argillite intercalated with thin to thick-bedded siltstone and fine-grained sandstone |
| | Tsuku Formation (Tu) | 25.3 | 18.0-30.0 | 27.0-30.0 | 10 | 1000 | Alternations of siltstone and argillite |
| Eocene | Chungling Formation (Cl) | 25 | 24.8-32.8 | 29 | 20 | 4000 | Argillite or slate, with thin bedded metasandstone |
| | Hsitsun Formation (Ht) | 25 | 22.2-32.6 | 30.5-33.5 | 10 | 2000 | Thin alternations of argillite and metasandstone |
| | Szeleng Sandstone (Em) | 23.5 | 18.1-32.0 | 28.0-32.0 | 10 | 2000 | Thick-bedded party pebbly quartzitic sandstone, arkosic sandstone and thin alternations, with argillite and thin coal seams on the upper part |

5

**Supplement of relationship of slope and landslide depth**

**Table S2  Relationship of slope and landslide depth (Chen et al., 2010)**

| Slope (degree) | Shimen watershed northern Taiwan (m) | Dajia watershed middle Taiwan (m) | Landslides triggered by Typhoon Aere in Shimen watershed northern Taiwan (m) | Average (m) | Classified Levels in Figure 8 (m) |
|---|---|---|---|---|---|
| <20 | 1.41 | 3.04 | 0.32 | 1.59 | 1.5 |
| 20~30 | 3.13 | 4.19 | 2.82 | 3.38 | 3.5 |
| 30~40 | 3.51 | 5.33 | 4.27 | 4.37 | 4.5 |
| 40~50 | 2.17 | 3.49 | 1.98 | 2.55 | 2.5 |
| >50 | 1.82 | 1.33 | 0.20 | 1.12 | 1.0 |

Chen S.C., Wu C.H., Wang Y.P. (2010) The Discussion of the Characteristic of Landslides Caused by Rainfall or Earthquake. Journal of Chinese Soil and Water Conservation, 41(2): 94-112. (in Chinese)

5

**Supplement of debris flow coverage area**

**Table S3  Debris flow coverage area for different use**

| Land Use Type | Scenario 1 (Square kilometer) | Scenario 2 (Square kilometer) |
|---|---|---|
| Transportation Loss | 0.22 | 0.25 |
| Household Loss | 0.09 | 0.10 |
| Publid Building  Loss | 0.03 | 0.03 |
| Forest Loss | 44.35 | 54.52 |
| Hydraulic Facility Loss | 1.70 | 1.89 |
| Industry and Commerce Loss | 0.07 | 0.07 |
| Agriculture Loss | 0.45 | 0.57 |

---

## Author Response (AR3)

**Reviewer 1: Anonymous**

**General comments**
This manuscript proposed the scenario approach coupled with landslide simulation, debris flow simulation and loss assessment. The reviewer believes that the approach to link the landslide simulation with debris flow simulation as well as loss assessment was unique and interesting but the manuscript should be improved to be published.
**Authors response:** We are grateful for the helpful specific comments. These comments should substantially improve the manuscript. Please see our response to the specific suggestions below.

**Specific comments**
**Reviewer 1:** In some parts of the manuscript, the sentences used by the authors are unclear. The authors are advised to rephrase these sentences in a better way or proof reading by English native speaker. One example is the sentence on Lines 9 to 13 of page 2.
**Authors:** We are sorry for the weak English description of this manuscript. The present version was modified by native speaker for better understanding and reading.

**Reviewer 1:** In line 7 of page 2, the authors mentioned that "some potential effects of landslides have been investigated by studying the differences between current and future scenarios". However, any detailed explanation about the potential effects was not provided, so the reviewer cannot understand the explanation.
**Authors:** We provided and explained the potential effects observed from past researches in line 25-28 of page 1.

**Reviewer 1:** In line 14 – 15 of page 2, the authors explained that the accumulated rainfall over a period of 3, 6, 12 hours exceeded the 200 year record. However, the authors did not provide exact rainfall amounts. To understand the climate condition of Taipei area, total amount of rainfall accumulations for 3, 6, 12 hour periods should be provided.
**Authors:** The maximum 3, 6, 12, and 72 hour rainfall during Typhoon Soudelor in Fushan meteorological station are 253mm, 442mm, 655mm, and 792mm respectively. The description was modified in line 12-13 of page 2.

**Reviewer 1:** In line 13 of page 4, "The TRIGRS is an inventory of shallow landslide simulation programs developed : : :.". But the TRIGRAS is not an inventory. Inventory means that the collection of past landslide features in a certain area for a certain period.
**Authors:** Thank you for your correction. We have rephrased the sentence in line 30 of page 3.

**Reviewer 1:** Does the subtitle "Landslide Inventory Simulation" mean that the location of past landslides

(That is, landslide inventory) were simulated and matched by TRIGRS? However, it does not seem that the analysis results of TRIGRS are matched to the location of past landslides in this manuscript. In this case, the term "landslide inventory simulation" should be revised.

**Authors:** Yes, this is what we mean. We have modified the title as "Shallow Landslides Simulation" to avoid misunderstanding in the present version.

**Reviewer 1:** In Fig. 3, the authors provided historical landslide area. The reviewer recommends to provide more detailed information about how the inventory was constructed and the landslide locations were obtained.

**Authors:** The historical landslide area (landslide inventory) were delineated by aerial photo by Central Geological Survey in Taiwan annually. The description was added in the caption of Fig. 3.

**Reviewer 1:** In 3.3, the soil parameters such as cohesion, friction angle, unit weight, hydraulic conductivity, and diffusivity were used as input values in landslide analysis using TRIGRS. However, any values for the soil parameters were not provided. Since the input parameters in TRIGRS are important, the values of the input parameters should be provided. In addition, since the soil thickness is also an important parameter affecting the simulation results, the detailed procedure to evaluate the soil thickness using slope-depth relationship should be explained.

**Authors:** The calibrated parameters of TRIGRS were provided in the present version, as shown in Table 3. The relationship of slope and landslide depth are based on the survey data in Taiwan (Chen et al, 2010), as shown in Table S1 (Supplement of relationship of slope and landslide depth).

**Reviewer 1:** The explanations in line 11 – 14 of page 13 is not clear. The reviewer cannot understand why and how calibration zones were reduced from 90 to 56. Please rewrite

**Authors:** There are 18 geologic settings and we set the landslide rate in 5 classes. Because zero landslide rate occurred in some geologic settings, only 56 zones were obtained for parameters calibration. We have rephrased our description in line 7-10 of page 12.

**Reviewer 1:** In line 7 – 8 of page 14, "Based on the landslide simulation results and soil thickness in each grid, the landslide inventory map were drawn, as depicted in Fig. 8, : : :". In reviewer's opinion, Fig. 8 is not landslide inventory but landslide analysis results. In addition, the author should provide clear explanation that the procedure that the authors performed and the results in Fig. 8 that the authors obtained. Since the most analysis results using TRIGRS show the distribution of factor of safety, the reviewer cannot understand the reason that Fig. 8 shows the soil depth as the results of analysis. Were the soil depths in Fig. 8 used in debris flow simulation? In the manuscript, any explanations were not provided.

**Authors:** Yes, the Fig. 8 is the TRIGRS simulation results. The soil thickness is one of the input for TRIGRS, and it was used for calculating the initial debris flow volumes in Debris-2D input. So we drew the TRIGRS simulation results with soil thickness in Fig. 8. We have added description in line 5-7 of page 13 and modified the caption of Fig. 8.

**Reviewer 1:** In line 7 – 11 of page 15, the reason that the concentration was presumed to be high and a maxima was used for the practical estimations should be explained. In addition, the meaning of the sentence

"the beginning of debris flow was assumed to be the same as the starting time." is not clear.

**Authors:** We have added some description to explain the calculation procedure in line 10-12 of page 7 and line 2-6 of page 16.

1﹒The equilibrium concentration of debris flow can be estimated by an empirical formula purposed by Takahashi (1981) in Eq. (3) and the maximum value cannot exceed 0.603 (Liu & Huang, 2003) which is occurred when the slope larger than $20.6^{\circ}$. Due to the slope of our most study basin even more than $20.6^{\circ}$, thus this paper direct take 0.603 for the concentration value of debris flow to estimate debris flow volumes in Eq. (2).

2﹒In reality, the landslide triggered debris flows in different locations could be occurred in different time. However, it is difficult to predict the debris flow occurred time after landslide, and in this paper what we concern is the final volume and influence area caused by debris flows. Therefore, the assumption of all the debris flows be the same starting time would not affect the final results.

**Reviewer 1:** In 4.2, the authors provides the calculation results of the possible economic losses using the quantified method in Table 1. However, the authors did not provide any values used in the calculations and the calculation procedures that the authors performed. The detailed information should be provided.

**Authors:** The calculation procedures of economic losses could be divided into three parts. First, we need to identify the impact area and depth of disaster which were got from the simulation results of debris flow. Then, the debris flow coverage area will be intersected with land-use map for identifying the loss of different use (e.g. household use, agriculture use, forest use etc.). Finally, the losses could be evaluated by loss functions and the corresponding parameters established in the database according to the uses. The total losses is the summation of the individual losses in different uses. The debris flow coverage area for different land use were provided in the Table 5. The calculation procedure was also provided in line 13-16 of page 7.

**Reviewer 2: Anonymous**

**General comments**

This manuscript describes an integrated approach to forecast the the economic impact of shallow landslides and debris flows in the framework of climate change scenarios. I personally think that the approach proposed is very interesting and useful since provides in quantitative terms the loss related to landslides and debris flows based on well-established literature methods. Anyway I think that the manuscript should be revised and improved before to be accepted for publication in the journal. In general the manuscript is well written but a revision of the manuscript structure and the clarification of some weak points would make the manuscript more clear and readable.

**Authors response:** We are grateful for the helpful specific comments. These comments should substantially improve the manuscript. Please see our response to the specific suggestions below.

**Specific comments**

**Reviewer 2:** I suggest you to revise the Methodology section. I think there is no need to describe in detail (with equations) the TRIGRS and Debris-2D models. In this section I would suggest you to explain better why you have selected these methods among all the literature ones and then refer to the original papers for further information about the model equations. Furthermore, at the beginning of sections 2.1, 2.2, 2.3 and 2.4 you provide a description of the state of the art. This parts should be moved in the Introduction. In general I suggest to shorten the methodology description, moving the state of the art in the Introduction, avoiding the description of the models and the subdivision in sub chapters (2.1, 2.2 and so on)

**Authors:** Thank you for your helpful suggestion. We have revised the introduction section and methodology section according to the suggestion. The description of state of the art have moved to the second paragraph of the introduction. The structure of methodology is also modified as two parts. The first section illustrate why we selected this models from past literatures and the second part illustrate how we integrate different models step by step. The suggestion structure substantially improve the manuscript.

**Reviewer 2:** In line1-2 of page 4 you state that the "spatial interpolation from 5 km to 40 m is made for the selected scenarios and used as inputs for landslide simulation." What do you mean for spatial interpolation? Please clarify and provide more information.

**Authors:** In TRIGRS simulation, the 40m×40m DEM was used as topography input. However, the spatial resolution of rainfall were in 5km×5km. To satisfy the spatial resolution as 40m×40m, the rainfall was interpolated by inverse distance weighting (IDW) method from 5km×5km to 40m×40m. We have rephrased the description in line 4-6 of page 6.

**Reviewer 2:** The sentence in line 13-14 of page 4 is not correct since TRIGRS is not an inventory of shallow landslides simulation but a physically-based model to forecast shallow landslides occurrence under rainfall events. Please rephrase

**Authors:** Thank you for your correction. We have rephrased the sentence according to the suggestion in line 30 of page 3.

**Reviewer 2:** The reviewer suggests to revise the term landslide in the methodology section. The landslides

simulated by the TRIGRS model are shallow landslides. I think you should use this term instead of the general term landslide which include all types of landslides.

**Authors:** Yes, you are right. We have revised the "landslides" as "shallow landslides" in the present version such as line 28, 31, 32 of page 6, line 2 of page 7, caption of 3.3, line 5-7 of page 13, and line 3 of page 16.

**Reviewer 2:** In Fig. 3 historical landslide area from 2008 and 2015 are reported. Please provide more information about how the inventory has been realized.

**Authors:** The historical landslide area (landslide inventory) were delineated by aerial photo by Central Geological Survey in Taiwan annually. We have modified the title of Fig. 8.

**Reviewer 2:** In line 4 of page 5 I suggest you to replace the term during with at the beginning.

**Authors:** Thank you for your suggestion. We have rephrased the term in line 25 of page 6.

**Reviewer 2:** In section 3.3 you don't provide any detailed information about the soil parameters used in the simulation of TRIGRS. In general in physically-based models the selection of soil parameters is an important issue. I suggest you to provide a table with soil parameters values and to describe how you have measured these data or which is the source.

**Authors:** Yes, the parameters are very important. The calibrated parameters of TRIGRS are provided in the Table 3. The calibrated procedure are described in line 12-24 of page 6. The hydraulic parameters were cited from past investigation (Central Geological Survey, 2010) and the soil parameters were calibrated by past events, detailed description is provided in section 3.3.

**Reviewer 2:** The sentence in line 11-14 at page 13 is not clear, please rephrase.

**Authors:** There are 18 geologic settings and we set the landslide rate in 5 classes, therefore totally we have 90 zones. However, there are some geologic settings are stable without landslides. So the total zones decrease to 56 zones for parameter calibration. We have rephrased the description in line 7-10 of page 12.

**Reviewer 2:** In my opinion the title of Figure 8 is uncorrect since the TRIGRS model provides factor of safety maps and not a map of soil depth. Please provide a figure with the results of the simulation and specify better what the figure 8 represents.

**Authors:** In Fig. 8, the shallow landslide area were simulated by TRIGRS. The soil depth were provided as reference because it is one of the input parameter of TRIGRS and DEBRIS-2D. We have rephrased the title of Fig. 8 and gave some description in line 5-7 of page 13 to avoid misunderstanding.

**Reviewer 2:** The results of loss assessment provided in section 4.2 are very interesting, anyway a clear explanation on how they have been obtained is missing. Please clarify better this point, providing clear description of calculation procedure.

**Authors:** Thank you for your suggestion. The calculation procedure have added in line 13-16 of page 7.

**Editor (Filippo Catani):**

Dear Authors

After reviewing the comments of referees and your replies, we believe that the manuscript is now almost ready for acceptance in NHESS. However, I have one minor, yet important, caveat on terminology. You repeatedly use the terms "soil depth" and "landslide depth" as if they were synonyms. They are not. This mistake may imply two options, which I ask you to clarify:

1. your model uses only landslide depths for modeling with TRIGRS or

2. your model has a separate soil depth modeling scheme (based only on slope angle which I do not personally endorse but which is acceptable) and then applies TRIGRS using this variable everywhere in the study areas.

In case 1 it is acceptable to mix the two terms but you are assuming that new landslides will have depths equal to the mapped ones, which is a little bit far fetched. In case 2 you may have a more general approach and better results for new landslides but you cannot mix the two terms. Please clarify this point and modify the manuscript accordingly (for example by avoiding the term "landslide depth"). After your new submission, the manuscript will only be revised by myself and then proceed to publication stage. I am sure you will be put forward this last effort to finalize a very nice paper.

best regards

FC

**Authors:**

Thank you for your positive response and the suggestion on the use of the terminology. As you mentioned in case 2, we used a separate soil depth based on slope angle in our model, and applied in TRIGRS and Debris-2D. The misuse "landslide depth" was avoided in the latest version of the supplement document. Besides, we also rephrased some sentences in line 11 of page 6, line 6 of page 13, and the title of Fig. 8.